# Efficient, Validation-Free Intrinsic Quality Estimation for Large-Scale Face Recognition Datasets

**Zhichao Chen** [* 1]  **Yongle Zhao** [* 1]  **Kaicheng Yang** [1]  **Meng Yang** [2]  **Yin Xie** [1]  **Ziyong Feng** [1]

## Abstract

We propose Intrinsic Quality (IQ), a validation-free metric designed to estimate the inherent potential of face recognition (FR) datasets to produce high-performance models without the need for full-scale training. IQ integrates two components: (i) a Neighbor-Consistency Score that quantifies local identity label agreement via nearest neighbors, and (ii) Global Representation Subspace Complexity (Effective Rank, ER), which captures the underlying embedding geometry and dataset diversity. IQ allows for rapid evaluation using lightweight proxy models or data subsets, facilitating dataset diagnosis and curation prior to resource-intensive full-scale training. We describe an experimental protocol tailored to clean, noisy, and mixed-quality FR datasets, and outline evaluation methodologies to validate IQ's predictive power for downstream performance.

## 1. Introduction

Deep face recognition (FR) has achieved remarkable progress in recent years, driven by advances in deep neural architectures, better optimization objectives, and the availability of large-scale training data. Modern FR systems are commonly trained with margin-based classification losses such as ArcFace (Deng et al., 2019), and benefit substantially from training sets containing millions of images and identities, including MS-Celeb-1M (Guo et al., 2016), VGGFace2 (Cao et al., 2018), and more recently the million-scale WebFace260M benchmark and its cleaned subset WebFace42M (Zhu et al., 2021). The latter explicitly demonstrates the practical importance of dataset scale and provides an automated self-training-based cleaning pipeline (CAST) to purify noisy web data, highlighting that large-scale FR training is increasingly *data-centric* rather than purely model-centric (Zhu et al., 2021). Recent facial representation pre-training methods further reflect this trend from a complementary direction, increasingly exploring how to exploit large amounts of unlabeled face data to reduce reliance on expensive identity annotations and improve representation quality (Zheng et al., 2022; Wang et al., 2023; Xie et al., 2025). In parallel, recent benchmarking efforts in broader vision-language settings further emphasize that dataset design and filtering can be treated as first-class research variables under standardized training recipes, enabling systematic dataset comparison and iteration (Gadre et al., 2023).

Despite these successes, constructing and maintaining high-quality large-scale FR datasets remains challenging. Weakly supervised web collection pipelines inevitably introduce closed-set label noise, identity ambiguity (merge/split), outliers, and heavy long-tail class imbalance. Such imperfections can degrade training stability and downstream accuracy, motivating extensive research on noisy-label learning and dataset cleaning for FR, including co-training and sample selection (e.g., Co-Mining (Wang et al., 2019)), graph-based label cleansing (e.g., Global-Local GCN (Zhang et al., 2020)), and meta-supervised cleaning (Zhang et al., 2021). While effective, these approaches predominantly act as *training-time* remedies and often require iterative optimization to validate dataset changes, which limits their use as fast, pre-training diagnostics for rapidly evolving datasets.

Assessing dataset quality *before* full-scale training is therefore critical in practice. Traditional evaluation relies on held-out validation splits with clean labels or repeated full training runs to measure downstream performance; both can be prohibitive in annotation cost, computation, and iteration time at modern scales. Moreover, clean validation splits are often unavailable in real-world FR due to privacy, licensing, and data-sharing constraints. More broadly, even widely used benchmarks may contain pervasive label errors that can destabilize evaluation and mislead dataset selection when presumed "ground truth" is imperfect (Northcutt et al., 2021b). Related work on dataset auditing and label issue identification (e.g., confident learning / Cleanlab) further suggests that label quality problems can be widespread

---

[*]Equal contribution [1]DeepGlint [2]School of Cyber Science and Technology, University of Science and Technology of China. Correspondence to: Ziyong Feng <ziyongfeng@deepglint.com>.

*Proceedings of the 43rd International Conference on Machine Learning*, Seoul, South Korea. PMLR 306, 2026. Copyright 2026 by the author(s).

and non-trivial even in standard settings (Northcutt et al., 2021a).

These constraints motivate *validation-free* dataset-intrinsic measures that can estimate whether a dataset variant is worth scaling *prior* to expensive training. In this paper, we use *trainability* to denote the downstream verification performance obtained by training with a *fixed* FR training and evaluation protocol on a given dataset variant. Our goal is not to predict the absolute best achievable performance under arbitrary architectures or hyperparameters, but to provide a practical intrinsic criterion for *ranking and monitoring* dataset variants under consistent settings.

Recent work in representation learning indicates that intrinsic properties of embeddings can predict downstream utility without labeled validation data. In self-supervised learning, RankMe shows that effective-rank-style spectral measures can serve as an unsupervised predictor of downstream performance and facilitate model selection without labels (Garrido et al., 2023); unsupervised embedding quality evaluation has also been explored through stability and separability criteria in high-dimensional geometry (Tsitsulin et al., 2023). In face analysis, a related but distinct line of work estimates *image-level* face quality without supervision, e.g., SER-FIQ (Terhorst et al., 2020) uses stochastic embedding robustness, while MagFace (Meng et al., 2021) learns embeddings whose magnitude correlates with image quality. These methods focus on sample-level suitability rather than dataset-level trainability under weak identity supervision. Meanwhile, broader data-centric diagnostics such as dataset cartography (Swayamdipta et al., 2020) and data valuation (Ghorbani & Zou, 2019) aim to understand or value training data, but often depend on repeated training or expensive attribution, making them less attractive for rapid iteration at FR scale.

In this work, we introduce *Intrinsic Quality* (IQ), a validation-free metric for large-scale face recognition datasets. IQ combines two complementary intrinsic signals derived from proxy embeddings: **Neighbor-Consistency** (Consis), which captures local label agreement via nearest-neighbor coherence, and **Global Representation Subspace Complexity**, quantified by (normalized) entropy-based effective rank. This combination is motivated by a key confounder in weakly supervised web data: global spectral complexity can increase under both beneficial data scaling and noise-induced complexity inflation, whereas neighborhood coherence tends to remain stable under clean scaling but degrades under corrupted supervision. This distinction is especially relevant in practice because web-collected FR datasets may become larger and noisier at the same time, requiring an intrinsic metric that remains informative in such mixed regimes. We validate trainability by measuring correlation and ranking consistency between intrinsic

scores computed from proxy embeddings and downstream verification performance under the standardized protocol.

Our contributions are summarized as follows:

- We propose IQ, a validation-free dataset-level metric for estimating the trainability of weakly supervised FR datasets using only the geometry of proxy embeddings.

- We empirically characterize common regimes in web-collected FR data—clean scaling, corrupted supervision, and their mixed setting—and show that combining local coherence with global complexity yields a consistent indicator across them.

- We provide an experimental protocol and analyze robustness with respect to proxy model capacity, sampling budgets, and comparisons with external validation-free baselines, enabling efficient estimation on million-scale datasets.

## 2. Related Work

### 2.1. Large-Scale Face Recognition Datasets and Intrinsic Noise

Large-scale FR performance is tightly coupled with the scale and coverage of training data. Widely used benchmarks such as MS-Celeb-1M (Guo et al., 2016), VG-GFace2 (Cao et al., 2018), and web-scale collections such as WebFace260M/WebFace42M (Zhu et al., 2021) have enabled substantial advances when paired with margin-based classification objectives (e.g., SphereFace (Liu et al., 2017), CosFace (Wang et al., 2018), ArcFace (Deng et al., 2019)). However, weakly supervised web pipelines inevitably introduce closed-set label noise, identity ambiguity (merge/split), outliers, and heavy long-tail imbalance, making it difficult to anticipate whether scaling will improve verification or primarily inject corruption. Beyond FR, recent evidence suggests that dataset "quality" can behave as an intrinsic factor not fully explained by size, balance, or architecture choice (Couch et al., 2025), further motivating explicit quality assessment.

### 2.2. Noisy-Label Learning and Dataset Cleaning for Face Recognition

A substantial body of work addresses label noise in FR via offline dataset cleaning or training-time robustness. Community and graph structure have been used to curate MS-Celeb-1M into cleaner subsets (Jin et al., 2018), and early large-scale FR pipelines explicitly considered noisy web supervision (Wu et al., 2018). Modern web-scale benchmarks provide scalable automated purification, e.g., CAST for WebFace260M (Zhu et al., 2021), demonstrating feasibility but also highlighting that residual ambiguity and

closed-set noise persist.

Beyond cleaning, many methods propose noise-aware training strategies. Co-Mining (Wang et al., 2019) uses co-training and sample exchange to mitigate noise, while Global-Local GCN (Zhang et al., 2020) leverages relational structure for large-scale label cleansing. Recent works more explicitly target closed-set noise and ambiguity through progressive correction or noise-adaptive mining, including BoundaryFace (Wu & Gong, 2022), RobustFace (Xin et al., 2024), and RepFace (Zhang et al., 2025). While effective, these approaches predominantly improve robustness during training and generally require substantial optimization to validate dataset changes, limiting their use as lightweight pre-training diagnostics.

### 2.3. Validation-Free Representation Diagnostics via Spectral Complexity and Dimensionality

A parallel line of work studies validation-free diagnostics using global statistics of representation geometry. Effective rank and related spectral summaries quantify the dispersion of singular values/eigenvalues and serve as proxies for effective dimensionality (Roy & Vetterli, 2007). In self-supervised learning, RankMe (Garrido et al., 2023) shows that effective-rank-style metrics correlate with downstream utility and can support model selection without labeled validation. Matrix information-theoretic analyses further motivate spectrum-based diagnostics by connecting covariance structure to entropy-like quantities (Zhang et al., 2023), and rank-based evaluation continues to develop in large models, e.g., Diff-eRank (Wei et al., 2024). Complementary to spectrum-only views, Lu et al. (2022) evaluate SSL representations via intrinsic dimension for expressiveness and a cluster-based learnability criterion, illustrating that combining global capacity indicators with local structure signals can reduce confounding effects.

### 2.4. Neighborhood Consistency and kNN-Based Reliability under Noisy Supervision

Local neighborhood structure provides a direct lens into semantic cohesion and label reliability. Deep k-Nearest Neighbors (DkNN) estimates prediction confidence by inspecting neighbor labels across layers (Papernot & McDaniel, 2018). Under label noise, kNN-style filtering and disagreement statistics can identify mislabeled points and improve learning; Bahri et al. (2020) provide empirical evidence and guarantees for deep kNN methods in noisy-label settings. Neighbor-consistency regularization further shows that encouraging agreement with nearest neighbors improves robustness under corrupted supervision (Iscen et al., 2022). Recent work also leverages strong embeddings to construct nearest-neighbor reliability scores for denoising at scale, e.g., WANN (Di Salvo et al., 2024). Together, these findings motivate neighborhood agreement as a label-coherence signal in weakly supervised FR datasets.

### 2.5. Low-Cost Predictors of Downstream Behavior

Related ideas have been explored in transfer learning, where one seeks to predict downstream performance without expensive fine-tuning. LEEP (Nguyen et al., 2020) and TransRate (Huang et al., 2022) are representative examples, and stability analyses highlight that such metrics can be sensitive to experimental choices, motivating protocol-aware evaluation (Agostinelli et al., 2022). Although these methods primarily target *model* transferability, they support a broader paradigm: low-cost intrinsic signals from representations can be informative about downstream outcomes under consistent settings.

**Summary.** Prior work suggests two useful but incomplete validation-free perspectives for web-collected FR: global spectral/dimensional summaries capture representation spread, while local neighborhood-based criteria reflect label coherence under noisy supervision. Since either perspective can be confounded in isolation (e.g., corruption inflating global complexity or clean scaling saturating local purity), it is natural to seek diagnostics that leverage complementary global and local signals for dataset-level assessment.

## 3. Methodology

### 3.1. Overview

Given a face recognition training set $\mathcal{D} = \{(x_i, y_i)\}_{i=1}^{N}$ with potentially noisy identity labels, we aim to estimate its *trainability* (as operationalized in Section 4.5) *without* access to any clean validation split. We propose **Intrinsic Quality (IQ)**, a dataset-level score computed from embeddings produced by a lightweight *proxy* model. IQ combines two complementary intrinsic signals: (i) a *local* signal, **Neighbor-Consistency** (**Consis**), measuring label agreement in $k$-NN neighborhoods; and (ii) a *global* signal, **Effective Rank** (**ER**), summarizing the spectrum of the embedding covariance. This design is motivated by a key confounder in weakly supervised web data: supervision corruption can reduce local coherence while inflating global spectral complexity, making either signal alone insufficient.

### 3.2. Proxy Embedding Extraction

We train a lightweight proxy FR model $f_\theta$ on $\mathcal{D}$ using a standard margin-based classification objective (e.g., Arc-Face (Deng et al., 2019)) and extract $d$-dimensional embeddings

$$e_i = f_\theta(x_i) \in \mathbb{R}^d, \qquad \|e_i\|_2 = 1. \qquad (1)$$

$\ell_2$ normalization makes cosine similarity equivalent to inner product, consistent with common angular-margin FR training (Deng et al., 2019).

**Identity-stratified sampling.** For very large $N$, we compute IQ on a subset $\widetilde{\mathcal{D}} \subset \mathcal{D}$ to control cost while preserving sensitivity to both diversity and noise. We sample approximately $M$ identities and draw $m$ images per identity (typically $m{=}10$), yielding $|\widetilde{\mathcal{D}}| = Mm$ embeddings. If near-duplicate images exist, we remove duplicates within each identity prior to sampling to avoid artificially inflating neighborhood agreement.

### 3.3. Neighbor-Consistency (Local Label Agreement)

Local neighborhood purity provides a direct lens into semantic cohesion and label reliability, and neighbor-consistency criteria are widely used in learning under noisy supervision (Iscen et al., 2022; Bahri et al., 2020). For each sampled embedding $e_i$, we retrieve its $k$ nearest neighbors $\mathcal{N}_k(i)$ under cosine similarity

$$\text{sim}(i,j) = e_i^\top e_j, \qquad (2)$$

excluding the trivial self-match. We define the per-sample neighborhood agreement

$$c_i = \frac{1}{k} \sum_{j \in \mathcal{N}_k(i)} \mathbf{1}\{y_j = y_i\}, \qquad (3)$$

and aggregate to a dataset-level statistic over the sampled subset

$$\bar{c} = \frac{1}{|\widetilde{\mathcal{D}}|} \sum_{i \in \widetilde{\mathcal{D}}} c_i. \qquad (4)$$

$\bar{c}$ is high when identities form locally label-consistent clusters, and decreases under label flips, identity merges/splits, or pervasive ambiguity.

### 3.4. Effective Rank (Global Subspace Complexity)

Let $E \in \mathbb{R}^{n \times d}$ be the embedding matrix for $n = |\widetilde{\mathcal{D}}|$ sampled points, with rows $e_i^\top$. We mean-center embeddings to remove a dominant mean direction:

$$\mu = \frac{1}{n} \sum_{i=1}^{n} e_i, \quad \widetilde{E} = E - \mathbf{1}\mu^\top. \qquad (5)$$

We compute the covariance

$$C = \frac{1}{n} \widetilde{E}^\top \widetilde{E} \in \mathbb{R}^{d \times d}, \qquad (6)$$

and obtain its eigenvalues $\{\lambda_\ell\}_{\ell=1}^d$. Following the spectral-entropy definition of effective rank (Roy & Vetterli, 2007),

$$p_\ell = \frac{\lambda_\ell}{\sum_{t=1}^d \lambda_t}, \qquad r_{\text{ent}} = \exp\left( -\sum_{\ell=1}^d p_\ell \log p_\ell \right). \qquad (7)$$

Intuitively, $r_{\text{ent}}$ increases when the spectrum spreads across more directions (richer subspace), but it may also increase when noise injects spurious variability.

### 3.5. Normalized Effective Rank

Because the entropy-based effective rank depends on the effective spectral support, for $n$ samples in $d$ dimensions it satisfies $1 \leq r_{\text{ent}} \leq Q$ where $Q = \min(n, d)$. To make ER comparable across settings with different $Q$, we use a log-normalized form:

$$\widetilde{r}_{\text{ent}} = \frac{\log r_{\text{ent}}}{\log Q}. \qquad (8)$$

We use natural logarithms; any base yields the same normalization if used consistently. The log mapping also compresses near-saturated regions of $r_{\text{ent}}$ and empirically yields more stable comparisons across dataset scales.

### 3.6. Intrinsic Quality (IQ)

We define IQ as a convex combination of local label coherence and normalized global complexity:

$$\text{IQ} = \alpha \cdot \bar{c} + \beta \cdot \widetilde{r}_{\text{ent}}, \qquad \alpha, \beta \geq 0, \ \alpha + \beta = 1. \qquad (9)$$

IQ is intended as an intrinsic proxy for dataset trainability: higher IQ should imply higher downstream verification performance under the standardized protocol in Section 4.5. The two terms are complementary in weakly supervised web data: $\bar{c}$ penalizes local incoherence induced by label noise or identity ambiguity (Iscen et al., 2022; Bahri et al., 2020), while $\widetilde{r}_{\text{ent}}$ captures how broadly embeddings span the representation subspace, reflecting diversity and representational richness (Roy & Vetterli, 2007; Garrido et al., 2023).

**Implementation note.** IQ requires proxy embeddings on $\widetilde{\mathcal{D}}$, exact $k$-NN retrieval for Consis (cosine similarity + top-$k$), and an eigen-decomposition of a $d \times d$ covariance matrix for $\widetilde{r}_{\text{ent}}$. With $|\widetilde{\mathcal{D}}| \approx 10{,}000$, exact $k$-NN and covariance-spectrum computation are already efficient, so no specialized nearest-neighbor acceleration is required.

## 4. Experimental Setup

### 4.1. Datasets and Noise Modeling

We conduct controlled studies on WebFace42M and two subsets with increasing scale: WebFace4M and WebFace12M (Zhu et al., 2021). We focus on the WebFace family because, to our knowledge, it is the largest public web-collected FR benchmark built under a weakly supervised collection-and-cleaning pipeline, making it particularly suitable for studying beneficial scaling versus supervision corruption within one consistent framework. These subsets enable isolating the effect of data scaling under a shared

collection and cleaning pipeline. To characterize robustness under corrupted supervision, we inject synthetic *closed-set* label noise on WebFace12M by randomly flipping identity labels at noise ratios $\{2\%, 5\%, 10\%, 20\%, 40\%\}$. Unless otherwise noted, "noise level 0" refers to the original labels of the corresponding dataset.

**Noise scope.** The injected noise is *uniform* label flipping and preserves the closed-set identity space. We use it as a controlled stress test to isolate one central FR failure mode— identity-label corruption—without conflating it with other dataset factors. This simplified setting is useful for disentangling the effects of scaling vs. corruption, but it is not intended to fully match real-world weak supervision. In addition to the pure-noise setting on WebFace12M, we also use these results to briefly examine a mixed regime in which increased scale and increased corruption are both present, by comparing a smaller clean set (WebFace4M) against a larger but progressively corrupted set (WebFace12M with injected noise).

### 4.2. Proxy Models and Embeddings

To compute intrinsic signals, we train lightweight proxy FR models and extract $\ell_2$-normalized embeddings from the final embedding layer. We use two commonly adopted backbone capacities to test robustness to proxy choice: ResNet-50 and ResNet-100 (He et al., 2015). Unless otherwise stated, intrinsic metrics reported in the main trend analyses use a fixed proxy configuration (ResNet-100, $d=1024$) to avoid confounding comparisons across datasets.

### 4.3. Sampling Protocol for Intrinsic Signals

To make intrinsic estimation feasible on million-scale datasets, we compute **Consis** and $\widetilde{r}_{\mathbf{ent}}$ on a sampled subset $\widetilde{\mathcal{D}}$. We use an identity-balanced sampling strategy: we first select approximately 1,000 identities and then sample 10 images per identity, resulting in roughly 10,000 images in total. When near-duplicate images exist within an identity, we deduplicate within the sampled pool to avoid artificially inflating neighborhood agreement.

Unless otherwise stated, we use cosine $k$-NN with neighborhood size $k=10$ and exclude the trivial self-match in neighbor retrieval.

### 4.4. IQ Instantiation and Recommended Weights

We instantiate IQ using Eq. (9) with fixed weights to avoid dataset-specific tuning. Unless otherwise specified, we use:

$$\alpha = 0.2, \qquad \beta = 0.8, \tag{10}$$

and keep $(\alpha, \beta)$ unchanged for all datasets, noise levels, and proxy backbones. This choice is motivated by the empirical behavior of the two components under our protocol. In

clean-scaling settings, Consis is often near-saturated and therefore provides limited dynamic range, whereas normalized effective rank typically varies more strongly and captures beneficial expansion of the embedding subspace. For this reason, we assign somewhat larger weight to $\widetilde{r}_{\mathrm{ent}}$. At the same time, ER alone is ambiguous because it can also increase under corrupted supervision, so Consis remains necessary as a corrective local term. Importantly, these weights are not tuned per dataset, and our conclusions do not depend on a narrowly optimized setting: in Section 5.4, we report a sensitivity analysis over $\beta$ (with $\alpha = 1 - \beta$) showing a broad high-correlation region rather than a sharp optimum.

### 4.5. Downstream Evaluation Protocol (Operationalizing Trainability)

To validate whether intrinsic signals reflect dataset *trainability*, we train full FR models on each dataset setting and evaluate downstream verification performance on the **MFR-ALL benchmark** (Huang et al., 2021) using standard verification accuracy. We treat this downstream performance under a fixed training and evaluation protocol as the empirical reference for trainability throughout the paper. All comparisons of intrinsic metrics are made against this consistent downstream protocol.

## 5. Evaluation and Analysis

### 5.1. Overview

We evaluate IQ primarily under two regimes that commonly arise in large-scale web-collected FR datasets: *clean scaling* (increasing dataset size under the same collection pipeline) and *label noise* (synthetic closed-set label flips). We report downstream verification performance on MFR-ALL (Section 4.5) together with intrinsic signals computed from proxy embeddings: **Consis** (Eq. (4)) and $\widetilde{r}_{\mathbf{ent}}$ (Eq. (8)). Unless otherwise specified, intrinsic signals are computed using a fixed proxy configuration (ResNet-100, 10k sampled embeddings, $d=1024$). We additionally include a targeted subset-selection experiment to assess IQ in a practical candidate-ranking scenario.

**Evaluation criterion (predictive validity).** We assess predictive validity by (i) correlation between intrinsic scores and downstream performance, and (ii) *ranking consistency* (whether intrinsic scores preserve the ordering of dataset settings under a fixed protocol). This matches our operational definition of dataset trainability in Section 4.5.

### 5.2. Core Trends

**Clean scaling.** We first study clean scaling from Web-Face4M to WebFace12M to WebFace42M. Table 1 summa-

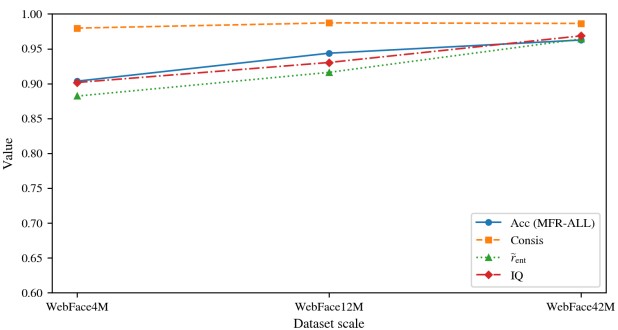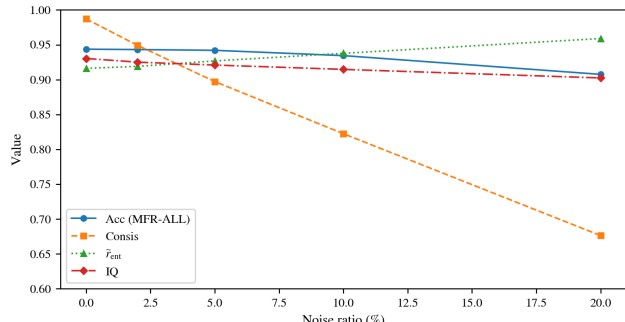

*Figure 1.* Trend summary under (left) clean scaling and (right) noise injection. Clean scaling increases $\widetilde{r}_{\text{ent}}$ while Consis remains near-saturated; noise increases $\widetilde{r}_{\text{ent}}$ but reduces Consis. IQ tracks downstream verification across both regimes.

*Table 1.* Effect of clean dataset scaling on downstream verification and intrinsic signals (proxy: ResNet-100; 10k samples; $d=1024$).

| Dataset | Acc | $\widetilde{r}_{\text{ent}}$ | Consis | IQ |
|---------|-----|------|--------|-----|
| WebFace4M | 90.36 | 0.882 | 0.980 | 0.902 |
| WebFace12M | 94.37 | 0.916 | 0.987 | 0.930 |
| WebFace42M | 96.26 | 0.964 | 0.986 | 0.968 |

rizes the results. As dataset size increases, downstream verification improves consistently. In parallel, $\widetilde{r}_{\text{ent}}$ increases, indicating that proxy embeddings span a broader global subspace as more data are introduced. In contrast, Consis remains near-saturated with only minor variation, suggesting that scaling mainly contributes *beneficial diversity* without substantially disrupting local label coherence.

**Label noise and a brief mixed-regime check.** Next, we inject synthetic closed-set label noise into WebFace12M. Table 2 shows that downstream verification degrades as noise increases. Notably, $\widetilde{r}_{\text{ent}}$ *also* increases with noise, demonstrating that global spectral complexity can be inflated by corrupted supervision. In contrast, Consis decreases substantially, reflecting a pronounced breakdown of local label agreement in embedding neighborhoods. The same table also provides a brief mixed-regime comparison by including WebFace4M (clean) alongside larger but noisier Web-Face12M variants. This confirms the intended role of IQ: although ER keeps increasing with added scale/noise, IQ tracks the downstream ordering more faithfully and can rank a smaller cleaner set above a larger but sufficiently corrupted one.

### 5.3. Diagnostics and Visual Evidence

**Trend summary: IQ separates scaling from corruption.** Figure 1 aggregates both regimes. A key observation is that $\widetilde{r}_{\text{ent}}$ rises in *both* scaling and noise settings, hence complexity alone is ambiguous. Consis remains near-saturated under

*Table 2.* Effect of synthetic label noise on verification and intrinsic signals, with a brief mixed-regime comparison (proxy: ResNet-100; 10k samples; $d=1024$). The WebFace12M rows isolate pure corruption, while comparison against clean WebFace4M illustrates the mixed scale–corruption case.

| Dataset | Noise (%) | Acc | $\widetilde{r}_{\text{ent}}$ | Consis | IQ |
|---------|-----------|-----|------|--------|-----|
| WebFace4M | 0 | 90.36 | 0.882 | 0.980 | 0.902 |
| WebFace12M | 0 | 94.37 | 0.916 | 0.987 | 0.930 |
| WebFace12M | 2 | 94.32 | 0.919 | 0.949 | 0.925 |
| WebFace12M | 5 | 94.21 | 0.927 | 0.897 | 0.921 |
| WebFace12M | 10 | 93.45 | 0.938 | 0.823 | 0.915 |
| WebFace12M | 20 | 90.76 | 0.959 | 0.676 | 0.903 |
| WebFace12M | 40 | 72.01 | 0.994 | 0.401 | 0.875 |

clean scaling but drops sharply under noise. IQ (Eq. (9) with fixed weights in Eq. (10)) combines these complementary behaviors and yields a consistent ordering that aligns with downstream verification across both regimes.

**Distributional view of neighborhood agreement.** Mean Consis can hide how neighborhood agreement changes across samples. We therefore inspect the distribution of per-sample agreement $c_i$ (Eq. (3)). Figure 2 visualizes the distribution across settings. Under clean scaling, $c_i$ remains concentrated near high agreement; under noise, the distribution shifts left and becomes more dispersed, indicating broad neighborhood disruption rather than a small number of isolated outliers.

**Spectral diagnostics beyond a single scalar.** Effective rank summarizes the eigen-spectrum but does not show *how* it changes. We therefore visualize (i) the log-eigenvalue spectrum and (ii) cumulative explained variance (CEV) in Figure 3. Clean scaling typically shifts the spectral elbow rightward and increases the number of components required to reach a fixed variance threshold, consistent with broader subspace coverage. Noise can also flatten the spectrum (complexity inflation), motivating the use of Consis to dis-

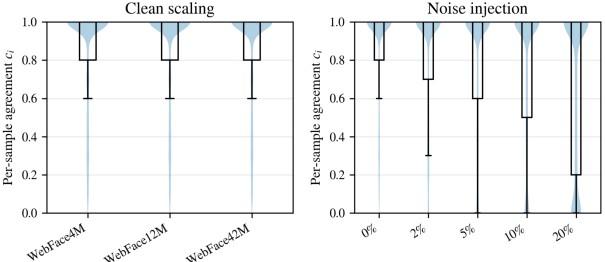

*Figure 2.* Distribution of per-sample neighbor agreement $c_i$ (Eq. (3)). **Left:** clean scaling maintains near-saturated neighborhoods with minor distributional change. **Right:** noise injection shifts the distribution toward lower $c_i$ values and increases dispersion, consistent with degraded local label coherence.

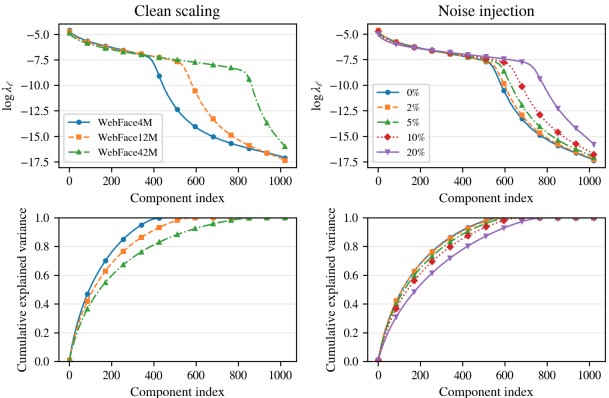

*Figure 3.* Spectral diagnostics. **Top:** log-eigenvalue spectra for scaling and noise settings; clean scaling shifts the elbow rightward, while noise can flatten the spectrum. **Bottom:** cumulative explained variance curves; clean scaling typically requires more components to reach the same variance threshold.

ambiguate beneficial diversity from corrupted supervision.

**Two-regime separation in the ($\widetilde{r}_{ent}$, Consis) plane.** Figure 4 plots settings in the 2D plane of ($\widetilde{r}_{ent}$, Consis). Clean scaling increases $\widetilde{r}_{ent}$ while maintaining high Consis; noise increases $\widetilde{r}_{ent}$ but simultaneously degrades Consis. This geometric separation explains why combining both signals yields a reliable dataset-level indicator.

## 5.4. Comparisons and Robustness

**Baselines: internal ablations and an external validation-free estimator.** The diagnostics above suggest that ER-only is confounded by noise-induced complexity inflation, while Consis-only can saturate under clean scaling. To position IQ relative not only to its own ablations but also to prior low-cost representation-quality estimators, we additionally compare against *RankMe* (Garrido et al., 2023), a represen-

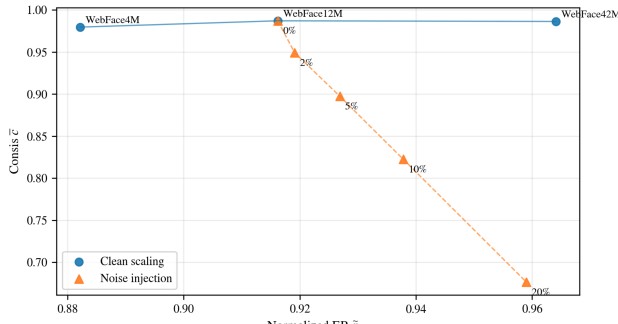

*Figure 4.* 2D visualization of settings in the plane of ($\widetilde{r}_{ent}$, Consis). Clean scaling and noise injection follow distinct trajectories: scaling increases global complexity while preserving local coherence; noise increases complexity while degrading coherence.

*Table 3.* Comparison with internal ablations and an external validation-free baseline on the union of scaling and noise settings. Correlation and ranking agreement are computed between each intrinsic metric and MFR-ALL accuracy. IQ uses Eq. (9) with fixed weights (Eq. (10)).

| Metric | Spearman | Pearson | Kendall $\tau$ |
|---|---|---|---|
| RankMe | 0.418 | 0.752 | 0.300 |
| ER-only ($\widetilde{r}_{ent}$) | 0.286 | 0.398 | 0.190 |
| Consis-only ($\overline{c}$) | 0.607 | 0.491 | 0.429 |
| IQ (ours) | 1.000 | 0.891 | 1.000 |

tative validation-free baseline based on effective-rank-style spectral complexity. We measure both correlation and ranking agreement with downstream accuracy across the union of scaling and noise settings. As shown in Table 3, RankMe improves over ER-only, confirming the utility of spectral complexity as a validation-free signal, but remains substantially below IQ. This is consistent with our motivation: spectral complexity alone is informative yet ambiguous in the presence of supervision corruption, whereas combining it with local label coherence yields a more reliable indicator.

**Targeted subset-selection experiment.** Although our main validation is intentionally controlled, IQ is intended for a practical use case: ranking candidate dataset variants before expensive full training. To strengthen this aspect, we construct three 12M-scale subsets within the Web-Face framework using identity-level intra-class feature variance: WebFace12M-HighVar, WebFace12M (original), and WebFace12M-LowVar. This provides a targeted subset-selection test within the same benchmark while preserving a consistent FR training and evaluation protocol. As shown in Table 4, IQ preserves the downstream ordering across these candidate subsets, supporting its intended role as a validation-free trainability proxy for dataset selection under fixed settings.

*Table 4.* Targeted subset-selection experiment within the WebFace framework (proxy: ResNet-100; 10k samples; $d=1024$). We construct three 12M-scale subsets using identity-level intra-class feature variance. IQ preserves the downstream ordering across candidate subsets, supporting its use for validation-free subset ranking under a fixed FR protocol.

| Dataset | Acc | $\widetilde{r}_{\mathrm{ent}}$ | Consis | IQ |
|---|---|---|---|---|
| WebFace12M | 94.37 | 0.916 | 0.987 | 0.930 |
| WebFace12M-HighVar | 94.45 | 0.915 | 0.996 | 0.932 |
| WebFace12M-LowVar | 93.04 | 0.896 | 0.982 | 0.913 |

*Table 5.* Stability under subset sampling (WebFace12M; proxy: ResNet-100; noise=0; $d=1024$).

| #Samples | $\widetilde{r}_{\mathrm{ent}}$ | Consis |
|---|---|---|
| 2k | 0.896 | 0.986 |
| 5k | 0.910 | 0.989 |
| 10k | 0.916 | 0.987 |
| 20k | 0.920 | 0.985 |
| 50k | 0.922 | 0.983 |
| 100k | 0.924 | 0.980 |

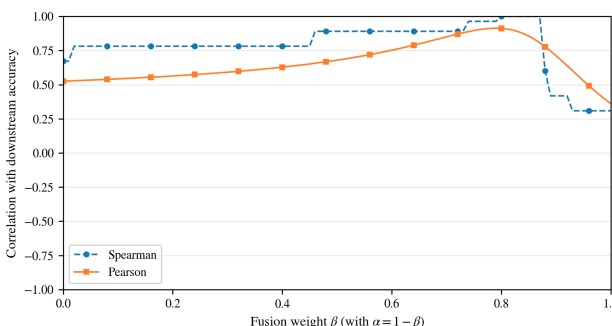

*Figure 5.* Sensitivity to fusion weight $\beta$. We sweep $\beta$ (with $\alpha = 1 - \beta$) and report correlation between IQ and downstream accuracy. A broad high-correlation region indicates robustness to moderate weight changes.

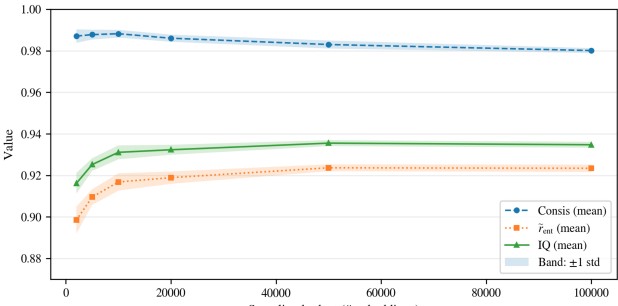

*Figure 6.* Sampling stability. Intrinsic estimates under varying sampling budgets, aggregated over repeated sampling seeds. Shaded regions indicate variability; estimates stabilize beyond a moderate budget.

**Sensitivity to the fusion weight $\beta$.** Although we fix $(\alpha, \beta)$ in the main experiments (Eq. (10)), this choice is intended as a stable default rather than a tuned optimum. We sweep $\beta$ (with $\alpha = 1 - \beta$) and report alignment between IQ and downstream accuracy. Figure 5 shows Spearman and Pearson correlation as a function of $\beta$, revealing a broad region with strong agreement. This supports the claim that IQ does not rely on a narrowly optimized weighting and that the main conclusions remain stable under moderate changes to the fusion weight.

**Sampling robustness and decision stability.** To ensure IQ is practical for million-scale datasets, we evaluate stability under different sampling budgets while holding the dataset and proxy fixed. Table 5 shows that both $\widetilde{r}_{\mathrm{ent}}$ and Consis stabilize beyond a moderate budget. To connect stability to the intended use case (ranking dataset variants), we further repeat sampling across random seeds and visualize mean±variability bands for Consis, $\widetilde{r}_{\mathrm{ent}}$, and IQ (Figure 6). In our setting, the resulting IQ ordering between major configurations remains stable across seeds once the sampling budget reaches the default regime.

**Robustness to proxy backbone choice (ranking consistency).** We test whether intrinsic *ordering* depends on proxy backbone capacity. Table 6 reports downstream accu-

racy and intrinsic signals under ResNet-50 vs. ResNet-100. While absolute intrinsic values can shift slightly with proxy strength, the qualitative trends and the relative ordering of dataset settings remain consistent.

## 6. Discussion and Conclusion

**What IQ measures (and what it does not).** IQ combines two complementary intrinsic signals—local neighborhood label coherence (Consis) and normalized global subspace complexity ($\widetilde{r}_{\mathrm{ent}}$)—to estimate dataset *trainability* without a clean validation set. Throughout this paper, trainability is operationalized as downstream verification performance under a fixed training and evaluation protocol (Section 4.5); consequently, IQ is intended to be used primarily for *ranking and monitoring* dataset variants under consistent settings, rather than predicting an absolute performance upper bound across arbitrary architectures or hyperparameter choices.

**Scope of the method.** IQ is not intended to be mathematically exclusive to face data. At a high level, it belongs to the broader family of validation-free representation diagnostics that combine global and local structure signals. What is specific here is the problem setting and instantiation: FR embeddings are typically $\ell_2$-normalized and compared in cosine space under angular-margin objectives, identity labels define a particularly strong local-neighborhood notion, and weakly supervised web FR datasets are strongly affected by

*Table 6.* Robustness to proxy backbone choice (10k samples; $d{=}1024$).

| Dataset | Proxy | Acc | $\widetilde{r}_{\mathrm{ent}}$ | Consis |
|---------|-------|-----|------|--------|
| WebFace4M | ResNet-50 | 86.33 | 0.878 | 0.973 |
| WebFace4M | ResNet-100 | 90.36 | 0.882 | 0.980 |
| WebFace12M | ResNet-50 | 91.16 | 0.907 | 0.982 |
| WebFace12M | ResNet-100 | 94.37 | 0.916 | 0.987 |
| WebFace42M | ResNet-50 | 93.03 | 0.954 | 0.981 |
| WebFace42M | ResNet-100 | 96.26 | 0.964 | 0.986 |

identity ambiguity and closed-set label corruption. In this sense, IQ is FR-motivated and FR-instantiated rather than only valid for faces.

**When and why IQ is useful.** Our experiments highlight a central confounder in weakly supervised web data: global complexity ($\widetilde{r}_{\mathrm{ent}}$) can increase under both beneficial scaling and supervision corruption, making complexity-only diagnostics ambiguous. This is also reflected in comparison with the external validation-free baseline RankMe (Garrido et al., 2023), which captures useful spectral information but remains limited when corruption inflates complexity. In contrast, Consis stays near-saturated when scaling mainly adds meaningful diversity, but degrades under label corruption and identity ambiguity. By fusing these signals with fixed weights (Eq. (9), Eq. (10)), IQ provides a practical intrinsic indicator that aligns with downstream verification across clean-scaling, pure-noise, mixed scale–corruption, and targeted subset-selection settings. Beyond a single scalar score, the distribution of per-sample agreement ($c_i$; Eq. (3)) can support targeted data debugging by showing whether quality issues are concentrated or broadly distributed. Finally, IQ does not rely on dataset-specific weight tuning: the fixed default weighting performs well across settings, and the $\beta$-sweep reveals a broad stable region rather than a sharp optimum.

**Practical considerations.** IQ is designed for fast iteration at scale. With identity-balanced subset sampling, exact $k$-NN and covariance-spectrum computation remain efficient, and the resulting intrinsic estimates are stable under repeated sampling seeds once the sampling budget reaches the default regime. In addition, trends and rankings remain qualitatively consistent across proxy backbone capacities, suggesting that IQ reflects dataset-intrinsic structure rather than artifacts of a specific proxy architecture.

**Limitations and future directions.** IQ has several important limitations. First, it depends on proxy embeddings; extremely weak proxies or substantial domain shift may reduce its reliability. Second, our controlled corruption uses uniform closed-set label flips, which provide a clean and interpretable stress test for identity-label corruption but do not fully capture realistic weak-supervision errors in large-

scale FR. Real web-collected datasets are often long-tailed and may contain more structured errors such as identity merge/split ambiguity, near-duplicate clusters, and structured confusion among visually similar identities. Accordingly, we view the current noise experiment as a controlled first step rather than a complete simulation of real-world dataset corruption. Third, while we evaluate trainability on MFR-ALL, the degree of alignment may vary with downstream benchmark composition and domain, so IQ should be interpreted as a validation-free proxy under a fixed FR protocol rather than a universal quality metric for all dataset settings. Future work should (i) evaluate IQ under more realistic structured-noise processes and dataset drift, (ii) study protocol sensitivity more systematically (e.g., different backbones, embedding dimensions, or training recipes), and (iii) explore closing the loop by using per-sample Consis diagnostics to guide automated cleaning and re-collection.

**Conclusion.** IQ offers a scalable, validation-free approach to assessing and monitoring large-scale face recognition datasets by leveraging the geometry of proxy embeddings. By combining global spectral complexity with local neighborhood coherence, IQ helps distinguish beneficial diversity from noise-induced complexity and supports rapid dataset iteration prior to resource-intensive full training.

## Impact Statement

This paper introduces a validation-free metric for estimating the trainability of large-scale face recognition datasets from proxy embedding geometry before expensive full-scale training. A potential positive impact is improved efficiency in data-centric face recognition research: IQ can help prioritize dataset variants, diagnose harmful label corruption, and support dataset cleaning or curation with reduced computation and annotation cost. In settings where clean validation splits are unavailable or expensive to obtain, such validation-free diagnostics may also make dataset iteration more practical and reproducible.

At the same time, face recognition is a sensitive application domain with significant ethical and societal risks. Methods that make it easier to screen, refine, or optimize face recognition training datasets could indirectly contribute to more capable biometric recognition systems, including in high-risk contexts such as surveillance, tracking, or other forms of identification. Moreover, a high IQ score does not imply that a dataset is fair, representative, privacy-preserving, or appropriate for deployment. We therefore view IQ as a tool for dataset diagnosis and comparison rather than a complete measure of dataset quality, and any practical use should be accompanied by additional assessment of fairness, demographic coverage, privacy, consent, licensing, legal compliance, and downstream task-specific behavior.

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
