# OpenReview forum: "Efficient, Validation-Free Intrinsic Quality Estimation for Large-Scale Face Recognition Datasets"
_ICML.cc/2026/Conference — ICML 2026 regular_

### Official Review · Reviewer_Updv · 2026-03-06

**Soundness:** 3
**Presentation:** 3
**Significance:** 2
**Originality:** 3
**Overall Recommendation:** 3
**Confidence:** 4

**Summary:**

The paper proposes a metric named Intrinsic Quality (IQ) to analyze the quality of face recognition datasets. The metric is defined as a weighted combination of two components: Neighbor Consistency and Effective Rank. Neighbor Consistency is designed to measure the correctness of identity labels in the dataset, while Effective Rank is used to evaluate the diversity of the dataset.
The paper shows that the increase in Effective Rank is consistent with the data scaling trend observed in the clean WebFace dataset, and that Neighbor Consistency can reflect the level of noise in the dataset. Several ablation studies are also conducted to analyze the proposed metric.

**Compliance With Llm Reviewing Policy:**

Affirmed.

**Final Justification:**

Thank you to the authors for their prompt and detailed responses during the rebuttal phase. I appreciate the clarifications provided.

After the rebuttal, I believe we have reached an agreement on several key points:

Experimental limitations:

(a) The experiments are conducted solely on WebFace-style datasets.

(b) The noise considered in the paper is simulated and limited to identity label flipping.


For (a), the authors have stated that they will revise the paper to explicitly narrow the scope of their method to datasets collected in a WebFace-like manner.
For (b), they have agreed to clearly acknowledge this limitation in the revised manuscript.

The remaining concern lies in the practical applicability of the proposed metric. Specifically, I am still unclear about its broader utility if it is only applicable to WebFace. This raises questions about its general significance and impact beyond this specific setting.

Overall, while I appreciate the authors’ efforts in addressing several concerns, this key issue remains unresolved. Therefore, I will slightly increase my score but maintain a negative overall assessment.

**Key Questions For Authors:**

1. The experiments are conducted only on the WebFace dataset. If IQ is intended to measure dataset quality in general, could the authors clarify why it is evaluated on only a single dataset?

2. The current experiments mainly consider identity label noise. However, dataset quality can also be affected by other factors, such as image blur, low resolution, or limited pose diversity within the same identity. Have the authors considered whether IQ can capture these types of quality issues？

3. Since the motivation is to support clean large-scale data collection, could the authors demonstrate a practical use case of IQ?

4. The paper suggests that Effective Rank tends to increase with data scaling, implying that maintaining high Neighbor Consistency is the key to preserving dataset quality. However, Neighbor Consistency is computed using KNN in the embedding space of a pretrained face recognition model. In this case, could a simpler strategy—such as computing the mean embedding for each identity and removing samples that fall outside the top-k similarity range—achieve similar dataset cleaning effects?

**Limitations:**

yes

**Strengths And Weaknesses:**

Strengths

1.	The paper is clearly written and easy to follow.

2.	The motivation of designing validation-free metrics for large-scale clean data collection is meaningful and relevant to practical dataset construction.

3.	The paper includes a relatively thorough ablation study analyzing the behavior of the proposed metric.
________________________________________

Weaknesses

1.	The experimental design is not sufficiently solid.

(a) The experiments are conducted only on the WebFace dataset. If IQ is intended to reflect dataset quality in general, it would be more convincing to evaluate it on multiple datasets constructed with different pipelines. This would better demonstrate the reliability and generalizability of the metric. Otherwise, it raises the question of whether IQ can only measure datasets generated with the same construction methodology.

(b) The noise analysis only considers identity label errors. However, other factors that significantly affect dataset quality—such as image blur, low resolution, or limited pose variation within an identity—are not considered. These factors could also influence the effective usefulness of the dataset for training face recognition models.

2.	The practical application scenario of the proposed method is not clearly demonstrated.

If the goal of the metric is to support clean large-scale data collection, it would be natural to demonstrate how IQ can be used to filter or rank data subsets. For example, one could sample multiple subsets (e.g., 12M images) from a larger dataset such as WebFace42M, rank them using IQ, and train models on these subsets to evaluate whether higher IQ scores indeed lead to better recognition performance. Such experiments would provide stronger evidence of the metric’s practical utility.

3.	The necessity of the proposed metric is not fully justified.

The paper suggests that data scaling naturally increases Effective Rank, implying that the key factor in maintaining dataset quality is ensuring that Neighbor Consistency does not decrease. However, Neighbor Consistency is computed based on KNN in the embedding space of a pretrained face recognition model. This raises the question of whether a simpler strategy—such as computing the mean embedding within each identity and removing samples that fall outside the top-k similarity range—could achieve similar dataset cleaning effects without requiring the IQ metric. The paper does not clearly demonstrate why the proposed metric is necessary beyond such straightforward filtering strategies.

---

> ### Author Rebuttal · Authors · 2026-03-30
>
> Thank you for the careful reading and constructive feedback. We agree that the current manuscript should better clarify the scope of IQ, its practical use case, and its distinction from instance-level filtering heuristics. We address these points below.
>
> **1. Why are the experiments conducted only on WebFace?**
>
> We agree that all current experiments are conducted on the WebFace family. However, our goal is narrower than a universal dataset-quality metric: IQ is intended as a **validation-free trainability proxy for large-scale weakly supervised face recognition datasets**, especially those obtained through web-scale collection and automatic cleaning. The WebFace benchmark is particularly suitable for this setting because it explicitly contains both large noisy web data and a cleaned subset produced by the CAST pipeline, making it well aligned with the problem studied here. We will revise the **title, abstract, and introduction** to make this scope explicit and avoid overstating generality.
>
> **2. Can IQ capture factors beyond identity-label noise?**
>
> Our current study focuses on **identity-supervision quality**, since identity inconsistency is one of the dominant failure modes in weakly supervised web-collected FR datasets. We do **not** claim that the current IQ formulation covers all aspects of “dataset quality.” That said, IQ is not conceptually limited to label flips: blur or low resolution can reduce embedding stability and local cohesion, while insufficient intra-identity variation can reduce effective subspace coverage. Prior work such as SER-FIQ also supports the view that image quality is reflected in embedding robustness. We will make this discussion explicit in the **limitations** section.
>
> **3. Practical use case: can IQ support dataset selection?**
>
> Yes. Following the reviewer’s suggestion, we added a targeted experiment to test IQ for **ranking candidate subsets before expensive full training**. Specifically, we construct three 12M-scale subsets using identity-level intra-class feature variance: **WebFace12M-HighVar**, **WebFace12M** (original), and **WebFace12M-LowVar**.
>   | Dataset | Acc(MFR-ALL) | $\widetilde r_{\mathrm{ent}}$ | $\textbf{Consis}$ | $\textbf{IQ}$ |
>   |---|---|---|---|---|
>   | WebFace12M | 94.37 | 0.916 | 0.987 | 0.930 |
>   | WebFace12M-HighVar | 94.45 | 0.915 | 0.996 | 0.932 |
>   | WebFace12M-LowVar | 93.04 | 0.896 | 0.982 | 0.913 |
>
> These results directly support the intended use case of IQ: the **IQ ranking is consistent with the downstream recognition ranking**. We will include this experiment in the revision as a practical validation of dataset selection.
>
> **4. Why is IQ needed beyond simpler rules such as removing samples far from the class mean?**
>
> We appreciate this question and agree the distinction should be made clearer. IQ is **not** intended to replace instance-level cleaning heuristics such as removing samples far from the class center. Those are sample-level filtering rules. IQ addresses a different problem: **dataset-level selection and ranking** before launching large-scale training. A class-center filter mainly removes outliers *within* an existing identity, but it does not provide a scalar criterion for comparing candidate datasets that may differ simultaneously in **noise, diversity, and scale**. IQ is designed to combine **global diversity** and **local consistency** into one dataset-level diagnostic. We will clarify this distinction more explicitly in the revision.
>
> **Planned revision.**
>
> We will revise the manuscript to:
> (i) narrow and clarify the scope of IQ as a trainability proxy for large-scale weakly supervised FR datasets;
> (ii) explicitly discuss non-label quality factors as limitations of the current study;
> (iii) add the new WebFace subset-selection experiment as a practical validation of IQ; and
> (iv) better distinguish dataset-level IQ from instance-level filtering heuristics.
>
> We thank the reviewer again for these helpful comments.

---

> > ### Author Rebuttal · Reviewer_Updv · 2026-03-31
> >
> > Thank you for the authors’ response. I still have the following two questions.
> >
> > First, regarding the experiments on the WebFace subset, I find them somewhat problematic. It is unclear how the high-variance and low-variance subsets are constructed. Why are only these two subsets selected?  Moreover, if we do not consider 𝑟_ent, we can still observe that higher **Consis** corresponds to higher accuracy. Does this suggest that subset selection could be performed without considering 𝑟_ent? Is it still necessary for IQ to incorporate 𝑟_ent? These concerns could potentially be addressed by providing clearer details on subset construction, as well as additional experiments with more randomly sampled subsets, along with correlation analysis between ACC and the metrics to support the claims.
> >
> > Second, in their response to the final question, the authors make a distinction between instance-level cleaning and dataset-level selection, arguing that IQ is designed for the latter. However, from an application perspective, if the goal is to obtain a better data subset, the roles of these two approaches appear similar. If the objective is instead to rank different datasets for model training (i.e., as a “trainability proxy”), then experiments across multiple datasets are necessary and cannot be avoided. However, as mentioned in the response to Q1, the authors do not plan to conduct such experiments, which makes it difficult for me to be convinced.

---

> > > ### Author Response · Authors · 2026-04-01
> > >
> > > Thank you for the follow-up questions. We clarify the two points below.
> > >
> > > **Q1: Construction of HighVar/LowVar and the necessity of $\boldsymbol{\widetilde r_{\mathrm{ent}}}$ .**
> > >
> > >   We apologize that the construction of **WebFace12M-HighVar** and **WebFace12M-LowVar** was not sufficiently clear. For each identity in WebFace42M, we compute pairwise feature distances among all samples and calculate the average variance of these distances. We then rank identities by this statistic: identities with the largest variance form **WebFace12M-HighVar**, and those with the smallest variance form **WebFace12M-LowVar**.
> > >
> > >  This design is motivated by the standard weakly supervised FR data construction pipeline: a pretrained face model first extracts embeddings from web images, and pseudo labels are then obtained by clustering. A **stricter clustering threshold** usually yields more compact identities (smaller intra-class variance), while a **looser threshold** yields larger intra-class variance and may also merge different real identities into one pseudo identity. In our paper, this latter effect is already simulated via **label-flip noise injection**. Since WebFace42M itself is relatively clean, **HighVar/LowVar** is used to simulate different levels of clustering-threshold strictness. We believe that combining **label noise simulation** and **intra-class variance variation** better approximates realistic weakly supervised FR data.
> > >
> > >  We also respectfully disagree that subset selection can rely on **Consis alone**. When label noise is limited, **Consis tends to saturate**, so improvements in dataset scale/diversity may not be reflected by Consis, although they still improve downstream performance. This is already shown in Table 1:
> > >   | Dataset | Acc(MFR-ALL) | $\boldsymbol{\widetilde r_{\mathrm{ent}}}$ | $\textbf{Consis}$ |
> > >   |---|---|---|---|
> > >   | WebFace4M | 90.36 | 0.882 | 0.980 |
> > >   | WebFace12M | 94.37 | 0.916 | 0.987 |
> > >   | WebFace42M | 96.26 | 0.964 | 0.986 |
> > >
> > >  From 4M to 42M, accuracy improves substantially, while **Consis changes only marginally**;  $\widetilde r_{\mathrm{ent}}$ , however, increases clearly. This is why IQ must include  $\widetilde r_{\mathrm{ent}}$ .
> > >
> > >  Moreover, Table 3 already shows that **Consis-only is weaker than IQ** across all studied settings:
> > >   | Metric | Spearman | Pearson | Kendall $\tau$ |
> > >   |---|---|---|---|
> > >   | ER-only( $\widetilde r_{\mathrm{ent}}$ ) | 0.286 | 0.398 | 0.190 |
> > >   | Consis-only( $\overline{c}$ ) | 0.607 | 0.491 | 0.429 |
> > >   | IQ(ours) | 1.000 | 0.891 | 1.000 |
> > >
> > >  Thus, neither ER-only nor Consis-only is sufficient; their combination is necessary.
> > >
> > > **Q2: Instance-level cleaning vs. dataset-level selection.**
> > >
> > >   These two problems are related but not equivalent. **Instance-level cleaning** mainly removes noisy samples within identities, which is close to the goal of **Consis**. In contrast, **IQ is a dataset-level measure** intended to jointly reflect **scale, diversity, and label cleanliness**, and ultimately correlate with the trainability of the whole dataset. A “cleaner” subset is not necessarily a better training set if it loses too much scale or diversity. Therefore, instance-level filtering cannot replace a dataset-level proxy.
> > >
> > >  We agree that validating IQ requires experiments across multiple dataset conditions. In fact, our experiments already cover three axes: **(1) dataset scale, (2) label noise, and (3) intra-identity diversity / variance**. We separately analyze IQ vs. downstream accuracy under these settings, and Table 3 summarizes the overall correlations. These results consistently support IQ.
> > >
> > >  We understand the concern that all experiments are based on the WebFace family. Our reason is that **WebFace42M is, to our knowledge, the largest publicly available weakly supervised web-scale FR dataset**, and thus is particularly suitable for the target setting of this paper. By controlled sampling from WebFace42M along the three axes above, we believe we can reasonably simulate the major variations encountered in large-scale weakly supervised FR datasets in practice. To avoid overstating our claim, we will revise the paper to clarify that IQ is intended as a **trainability proxy for large-scale weakly supervised FR datasets in the WebFace-style pipeline**, rather than a universally validated metric for arbitrary face datasets.
> > >
> > >  Finally, we sincerely thank the reviewer again for the constructive feedback. If the above clarifications address the remaining concerns, we would greatly appreciate it if the reviewer could consider raising the overall recommendation score accordingly.

---

### Official Review · Reviewer_NZwp · 2026-03-11

**Soundness:** 3
**Presentation:** 3
**Significance:** 3
**Originality:** 2
**Overall Recommendation:** 4
**Confidence:** 2

**Summary:**

This paper studies validation-free dataset quality estimation for large-scale face recognition. The main idea is to estimate a dataset’s “trainability” without relying on a clean validation split or repeated full training runs. The proposed metric, Intrinsic Quality (IQ), combines a local signal—Neighbor-Consistency (the fraction of same-label nearest neighbors in proxy embedding space)—with a global signal—normalized effective rank of the embedding covariance. The authors argue that this combination helps distinguish beneficial data scaling from harmful supervision noise, because effective rank alone can increase in both cases while local label consistency tends to drop under corruption.

**Compliance With Llm Reviewing Policy:**

Affirmed.

**Final Justification:**

Thanks for your rebuttal. My concern has been addressed.

**Key Questions For Authors:**

See weakness please.

**Limitations:**

Yes. The author says there is no limitations.

**Strengths And Weaknesses:**

1. Strength
   1. This paper provides a novel metric named IQ for diagnosis before expensive large-scale face recognition training. This is useful for data-centric iteration, especially when clean validation labels are unavailable.
   2. The presentation of the central idea is mostly clear and easy to understand.
2. Weakness
   1. The motivation is hard to understand. Why did the author choose Neighbor-Consistency and normalized effective rank as the metrics? What is the merit of these two metrics for dataset evaluation?
   2. The empirical validation is quite narrow. The paper effectively evaluates only a handful of dataset conditions.
   3. IQ does not seem to be a special metric for the face dataset. Are there any special designs for the face dataset, or is it just a common metric that can apply to any datasets?

---

> ### Author Rebuttal · Authors · 2026-03-30
>
> We thank the reviewer for the careful reading and constructive feedback. We agree that the current version should better explain the motivation of the two components, clarify the scope of IQ, and strengthen the empirical validation.
>
> **1. Why Neighbor-Consistency and normalized Effective Rank?**
>
> Our design is motivated by a key confounder in large-scale weakly supervised FR: under clean scaling, representation complexity increases, but under corrupted supervision, spectral complexity can also increase. Therefore, a global complexity statistic alone is ambiguous. Effective-rank-style measures are attractive because they provide a label-free summary of how broadly embeddings occupy the representation subspace, and prior work such as RankMe shows that effective rank can correlate with downstream utility.
>
> Neighbor-Consistency is the complementary local signal. FR embeddings are explicitly trained in cosine/angular space; margin-based objectives such as ArcFace use normalized features where local same-identity neighbor agreement is a natural measure of label coherence. Thus, the two terms capture different failure modes: ER measures global representational coverage, while Consis measures local label coherence. Our claim is not that these are the only possible choices, but that they form a simple and effective complementary pair for the trainability-ranking problem studied here. We will revise Sec. 1 and Sec. 3 to make this motivation more explicit.
>
> **2. The empirical validation is narrow.**
>
> We agree that the current validation is controlled rather than broad. We focus on the WebFace family because it is, to our knowledge, the largest public web-collected FR benchmark constructed under a weakly supervised collection-and-cleaning pipeline, making it particularly suitable for studying beneficial scaling versus supervision corruption within one consistent framework.
>
> To strengthen the practical evidence, we added a targeted subset-selection experiment on WebFace. Our rationale is that different sampling strategies within this benchmark can simulate candidate datasets with different characteristics. Specifically, we construct three 12M-scale subsets using identity-level intra-class feature variance: **WebFace12M-HighVar**, **WebFace12M** (original), and **WebFace12M-LowVar**.
>
> Results:
>   | Dataset | Acc(MFR-ALL) | $\widetilde r_{\mathrm{ent}}$ | $\textbf{Consis}$ | $\textbf{IQ}$ |
>   |---|---|---|---|---|
>   | WebFace12M | 94.37 | 0.916 | 0.987 | 0.930 |
>   | WebFace12M-HighVar | 94.45 | 0.915 | 0.996 | 0.932 |
>   | WebFace12M-LowVar | 93.04 | 0.896 | 0.982 | 0.913 |
>
> This directly supports the intended use case of IQ: **ranking candidate subsets before expensive full training**, where the IQ ordering is consistent with the downstream recognition ordering. We will include this experiment in the revision. We will also clarify that IQ is intended as a **validation-free trainability proxy under a fixed FR protocol**, rather than a universal quality metric for all dataset settings.
>
> **3. Is IQ really specific to face datasets?**
>
> IQ is not claimed to be mathematically exclusive to face data. At a high level, it belongs to the broader family of validation-free representation diagnostics. What is specific here is the **problem setting and instantiation**: FR embeddings are typically $\ell_2$ -normalized and compared in cosine space under angular-margin objectives; identity labels define a particularly strong local-neighborhood notion; and weakly supervised web FR datasets are dominated by identity ambiguity and closed-set label corruption, where neighborhood label agreement is especially meaningful. Thus, our position is that IQ is **FR-motivated and FR-instantiated**, rather than “only valid for faces.” We will revise the manuscript to make this scope clearer.
>
> **4. Clarification on limitations.**
>
> We apologize if the limitations were not stated clearly enough. We do not intend to claim that the method has no limitations. The main limitations are: dependence on proxy embeddings, controlled synthetic noise being simpler than real-world merge/split errors, and possible protocol sensitivity across downstream benchmarks. We will revise the limitations section to make these points explicit.
>
> **Planned revision.**
>
> We will revise the manuscript to:
> (i) better motivate why ER and Consis are complementary in weakly supervised FR;
> (ii) narrow and clarify the scope of the claim;
> (iii) add the new WebFace subset-selection experiment as a practical validation of IQ; and
> (iv) strengthen the limitations discussion.
>
> We thank the reviewer again for these helpful comments.

---

> > ### Author Rebuttal · Reviewer_NZwp · 2026-04-05
> >
> > Thanks for your rebuttal. My concern has been addressed.

---

### Official Review · Reviewer_nNRQ · 2026-03-12

**Soundness:** 3
**Presentation:** 3
**Significance:** 3
**Originality:** 3
**Overall Recommendation:** 4
**Confidence:** 3

**Summary:**

This paper proposes an unverified set-based metric called IQ for evaluating the quality of large-scale face recognition datasets. The authors discovered that the expansion of the dataset and the increase of label noise would lead to an increase in the complexity of the global representation subspace. To address this, the authors introduce local neighbor consistency to penalize the noise. The paper has a clear structure and a clear motivation, and has significant implications for the practical industrial and academic communities.

**Compliance With Llm Reviewing Policy:**

Affirmed.

**Final Justification:**

My problem has been solved.

**Key Questions For Authors:**

Please see the Strengths And Weaknesses.

**Limitations:**

Yes.

**Strengths And Weaknesses:**

This paper has a very good motivation, and there are several questions that need to be answered:
1、If the dataset itself is filled with significant noise, then the proxy model trained on this dataset will be poor. Would it be even more unlikely to calculate the credibility of kNN then?

2、For formula 10, what is the basis for choosing the two variables as 0.2 and 0.8?

3、The noise injection experiment in the paper utilizes closed-set label flipping. However, in real large-scale face datasets, the data is uneven. Is this setup feasible?

---

> ### Author Rebuttal · Authors · 2026-03-30
>
> Thank you for the positive assessment and for the constructive questions. We are encouraged that the reviewer finds the motivation clear and the contribution meaningful. We address the three points below.
>
> **1. If the dataset is very noisy, would the proxy model become too poor for reliable kNN?**
>
> This is an important concern. We agree that when the dataset itself is highly noisy, a proxy model trained on it will also be weaker. However, this is still consistent with our goal. IQ is **not** intended to rely on a strong full-scale model; instead, it uses a lightweight proxy only to provide a **coarse embedding geometry** for relative dataset evaluation. In face recognition, ArcFace-style training produces ell_2-normalized embeddings with cosine/angular neighborhood structure, so even a moderate proxy can still provide meaningful local geometry.
>
> Importantly, if a dataset is heavily corrupted, the proxy embedding quality also degrades, which naturally leads to lower Neighbor-Consistency and thus a lower IQ score. This is aligned with the purpose of IQ: to reflect dataset trainability rather than to build a strong recognition model. Empirically, our robustness study already shows consistent trends across proxy capacities (ResNet-50 vs. ResNet-100), and in additional experiments we simulated label-flip noise up to 40%; the resulting IQ ranking remains highly consistent with the final downstream accuracy ranking. We will clarify in the revision that IQ is intended for **relative ranking of dataset variants under a fixed proxy/training recipe**, not for absolute estimation under arbitrarily severe corruption.
>
> **2. Why choose **$\alpha=0.2$**, **$\beta=0.8$** in Eq. 10?**
>
> The weights were not tuned per dataset. We chose a fixed setting to avoid overfitting and because the two components operate in different regimes: in our experiments, Consis is often near-saturated under clean scaling, while normalized ER carries more variation across beneficial scaling; hence we assign somewhat larger weight to ER. At the same time, ER alone is ambiguous because it can also increase under noise, so Consis remains necessary as the corrective local term.
>
> We agree this point should be explained more clearly. We already include a β-sweep showing a broad high-correlation region rather than a sharp optimum(Fig.5), and in the revision we will make this motivation explicit and emphasize that the main conclusion does **not** rely on a narrowly tuned (0.2, 0.8) choice.
>
> **3. Is closed-set label flipping realistic given that real large-scale face data are imbalanced and noisy in more complex ways?**
>
> We agree that uniform closed-set flipping is a simplified stress test. We used it because it is controlled and lets us isolate one central FR failure mode—identity-label corruption—without conflating it with other dataset factors. This is also consistent with prior FR work that explicitly studies closed-set noise as a major challenge in web-collected face data.
>
> That said, we do not claim that this fully matches real-world weak supervision. Real large-scale FR datasets are indeed long-tailed and contain more structured errors such as merge/split ambiguity and duplicate clusters. The WebFace260M/WebFace42M benchmark itself was introduced in precisely the context of noisy web-collected FR data and automated cleaning via CAST.   In the revision, we will make this limitation more explicit and position the current experiment as a controlled first step, while discussing more realistic structured-noise settings as future work.
>
> **Planned revision.**
>
> We will revise the paper to:
> (i) better explain why a lightweight proxy is sufficient for relative ranking;
> (ii) clarify the rationale for fixed IQ weights and highlight the weight-sensitivity results; and
> (iii) state more explicitly that closed-set flipping is a controlled simplification rather than a full model of real FR noise.
>
> Thank you again for the helpful comments.

---

> > ### Author Rebuttal · Reviewer_nNRQ · 2026-04-04
> >
> > My problem has been solved. I hope the author can make thorough revisions in the revised version. I will also keep my attention  on the paper.

---

### Official Review · Reviewer_KbzK · 2026-03-12

**Soundness:** 3
**Presentation:** 3
**Significance:** 2
**Originality:** 3
**Overall Recommendation:** 5
**Confidence:** 3

**Summary:**

This paper addresses the problem of estimating the trainability of large-scale, weakly supervised face recognition (FR) datasets without requiring expensive full-scale training runs or clean validation splits. The authors propose Intrinsic Quality (IQ), a dataset-level metric computed from the embedding geometry of a lightweight proxy FR model trained on a sampled subset of the data. IQ integrates two complementary components: Consis, which measures local label agreement via nearest-neighbor coherence in embedding space, and Effective Rank (ER), a global entropy-based measure of how broadly the embedding subspace is occupied.
The metric is designed to support rapid dataset diagnosis and curation prior to resource-intensive full training. The authors evaluate IQ under two common regimes--clean scaling (increasing dataset size under a fixed pipeline) and corrupted supervision (synthetic closed-set label noise injection)--using public WebFace datasets. Results show that Consis and ER exhibit complementary behaviors across these regimes, and that their combination as IQ yields stronger and more consistent correlation with downstream verification performance than either component alone. Additional experiments validate sampling stability across varying budget sizes and robustness to the choice of fusion weight.

**Compliance With Llm Reviewing Policy:**

Affirmed.

**Final Justification:**

The rebuttal addressed all my concerns adequately, and I see the current form of the paper as a significant contribution.

**Key Questions For Authors:**

1. The ER metric rises under both clean scaling and corrupted supervision. What happens in the regime where both effects are present simultaneously—a large but heavily corrupted dataset? Does IQ still reliably rank such a dataset below a smaller, cleaner one?
2. The caption on fig. 2 references the “left” shift of the distribution, but the direction appears inconsistent with the graph layout as presented. Could the authors confirm and correct the labeling?
3. Given the stated broad ML motivation, have the authors considered whether IQ generalizes beyond face recognition to other weakly supervised recognition tasks (e.g., person re-identification, product retrieval)? Even a brief qualitative discussion of expected transferability would strengthen the significance of the contribution.

**Limitations:**

The limitations section acknowledges that IQ is intended for ranking under consistent settings rather than absolute performance prediction, and that future work should evaluate under more realistic noise processes. However, the paper does not discuss broader societal implications of a tool designed to improve face recognition dataset quality. Advances in FR dataset curation have dual-use implications: improving surveillance capabilities alongside legitimate identity verification applications. A brief acknowledgment of these downstream risks would be appropriate, particularly given the venue's ethics expectations.

**Strengths And Weaknesses:**

Strengths:
- The core experimental setup is well-designed. Injecting synthetic label noise (closed-set identity flips at varying rates on WebFace12M) isolates the supervision quality signal cleanly from scale effects, making the ablation easy to interpret.
- The dual-metric logic is convincing: ER alone is ambiguous because it rises under both clean scaling and corrupted supervision, so it can’t distinguish the two. Consis breaks the tie. The correlation analysis makes this concrete--ER-only shows weak association with downstream performance, Consis-only improves on that but is still imperfect, and IQ is consistently the best of the three. That’s a clean, honest argument for the combination.
- The metric is actionable. Low Consis points to label noise; low ER points to insufficient diversity. That distinction is useful for curators who need to know where to focus cleaning effort.
- Related work is thorough and honest about the lineage of the method--the paper traces how prior work on cluster-based learnability and neighborhood coherence motivated IQ without overclaiming novelty. All datasets used (WebFace4M, WebFace12M, WebFace42M) are public, which helps reproducibility.

Weaknesses:
- There is no experimental comparison against existing metrics that serve a similar purpose. The paper compares IQ’s two components against each other (ER-only vs. Consis-only vs. IQ), but does not pit IQ against any prior dataset quality or trainability estimator from the literature. Without at least one external baseline, it’s hard to know whether IQ is better than what already exists or simply better than its own ablations.
- The ER metric is acknowledged to be confounded, but the failure modes of the combined IQ score are not discussed. What happens when a dataset is both very large and heavily corrupted--where scaling and noise effects are present simultaneously? It’s not obvious that IQ would still rank it correctly relative to a smaller, cleaner dataset.
- The paper’s impact statement frames IQ as a contribution to broad ML, but every experiment is in face recognition. There’s no discussion of whether the metric would transfer to other weakly supervised recognition tasks like person re-identification or product retrieval, where similar noise and curation challenges arise. The broad ML claim needs either evidence or a more honest scope.

---

> ### Author Rebuttal · Authors · 2026-03-30
>
> Thank you for the thoughtful and positive assessment. We appreciate the reviewer’s recognition of the controlled setup, the complementary logic of ER and Consis, and the practical usefulness of the metric. We address the main questions below.
>
> **1) Comparison against external baselines.**
>
> We agree this is an important omission in the current version. Our original comparisons (ER-only / Consis-only / IQ) establish complementarity, but do not yet position IQ against prior low-cost representation-quality estimators. To address this, we added **RankMe** as an external baseline. RankMe is particularly relevant because it uses effective rank as a validation-free predictor of downstream utility in representation learning.   The results are:
>   | Metric | Spearman | Pearson | Kendall $\tau$ |
>   |---|---|---|---|
>   | RankMe | 0.418 | 0.752 | 0.300 |
>   | ER-only( $\widetilde r_{\mathrm{ent}}$ ) | 0.286 | 0.398 | 0.190 |
>   | Consis-only( $\overline{c}$ ) | 0.607 | 0.491 | 0.429 |
>   | IQ(ours) | 1.000 | 0.891 | 1.000 |
>
> These results show that IQ improves not only over its own ablations, but also over a strong external baseline based on effective rank alone.
>
> **2) What happens when scaling and corruption are both present?**
>
> This is an excellent question. Our original experiments separated the two regimes intentionally because the WebFace benchmark was designed to expose the contrast between noisy large-scale web data and cleaned data under the CAST pipeline, enabling controlled analysis.   We agree, however, that the mixed regime is an important stress test. We therefore added the following experiment:
>   | Dataset | Noise(%) | Acc(MFR-ALL) | $\widetilde r_{\mathrm{ent}}$ | $\textbf{Consis}$ | $\textbf{IQ}$ |
>   |---|---|---|---|---|---|
>   | WebFace4M | 0 | 90.36 | 0.882 | 0.980 | 0.902 |
>   | WebFace12M | 0 | 94.37 | 0.916 | 0.987 | 0.930 |
>   | WebFace12M | 10 | 93.45 | 0.938 | 0.823 | 0.915 |
>   | WebFace12M | 20 | 90.76 | 0.959 | 0.676 | 0.903 |
>   | WebFace12M | 40 | 72.01 | 0.994 | 0.401 | 0.875 |
>
> This result supports the intended role of IQ in the mixed regime: although ER keeps increasing with added scale/noise, Consis decreases substantially, and the combined IQ score tracks the downstream ranking much better than ER alone. In particular, IQ correctly ranks a smaller cleaner set above a larger but sufficiently corrupted one.
>
> **3) Scope beyond face recognition / broad-ML framing.**
>
> We agree that the original wording can overstate the scope. IQ is **not** claimed to be universally validated across all ML settings; rather, it is **FR-motivated and FR-instantiated**. This is important because FR embeddings are typically ell_2-normalized and compared in cosine/angular space under objectives such as ArcFace, making local neighbor agreement especially meaningful in this domain.   At a higher level, IQ belongs to the broader family of validation-free representation diagnostics, but the current paper demonstrates it only in face recognition. We will revise the impact/scope language accordingly and avoid overclaiming transfer beyond FR.
>
> **4) Figure 2 caption.**
>
> Thank you for catching this. We will check the “left shift” wording against the final figure layout and correct the caption if needed.
>
> **Planned revision.**
>
> In revision, we will:
> (i) add RankMe as an explicit external baseline;
> (ii) include the new joint scale+corruption stress test;
> (iii) narrow the broad-ML wording and clarify the FR-specific scope; and
> (iv) fix the Figure 2 caption inconsistency.
>
> Thank you again for the constructive suggestions.

---

> > ### Author Rebuttal · Reviewer_KbzK · 2026-04-02
> >
> > The authors addressed all three of my main concerns directly. They added RankMe as an external baseline, and the results clearly show IQ outperforming it -- closing the most significant gap in the original evaluation. The new joint scale+corruption experiment confirms that IQ correctly ranks datasets in the mixed regime, with Consis compensating for ER's ambiguity as expected. The authors also committed to narrowing the broad-ML scope language, which was the right call. The Figure 2 caption issue was acknowledged and will be corrected. I am satisfied with the rebuttal and will adjust my recommendation to Accept.

---

### Decision · Program_Chairs · 2026-04-30

**Decision:**

Accept (regular)

**Comment:**

The paper received mixed recommendations, with three reviewers supporting acceptance and one reviewer maintaining a weak reject. Reviewers recognized the practical value of a validation-free metric for estimating the trainability of large-scale face recognition datasets, and found the combination of neighbor consistency and effective rank well motivated. After rebuttal, the main concerns were addressed to the satisfaction of the positive reviewers. The main remaining concern was the boundary of applicability, namely whether the current evidence is sufficient to justify significance beyond WebFace-style weakly supervised face recognition datasets. AC agrees that the scope should be stated more carefully in the final version, but does not view this remaining concern as sufficient to outweigh the paper's utility and relevance in an important practical setting. AC therefore recommends acceptance. In the revision, the authors should explicitly narrow the scope claims, strengthen the discussion of limitations and broader implications, and incorporate the rebuttal clarifications and additional experiments into the final paper.